# Corrosion-resistant cobalt phosphide electrocatalysts for salinity tolerance hydrogen evolution

Xinwu Xu[1,3], Yang Lu[1,3], Junqin Shi[1] ✉, Xiaoyu Hao[1], Zelin Ma[1], Ke Yang[1], Tianyi Zhang[1], Chan Li[1], Dina Zhang[1], Xiaolei Huang[2] ✉ & Yibo He[1] ✉

Seawater electrolysis is a viable method for producing hydrogen on a large scale and low-cost. However, the catalyst activity during the seawater splitting process will dramatically degrade as salt concentrations increasing. Herein, CoP is discovered that could reject chloride ions far from catalyst in electrolyte based on molecular dynamic simulation. Thus, a binder-free electrode is designed and constructed by in-situ growth of homogeneous CoP on rGO nanosheets wrapped around the surface of Ti fiber felt for seawater splitting. As expected, the as-obtained CoP/rGO@Ti electrode exhibits good catalytic activity and stability in alkaline electrolyte. Especially, benefitting from the highly effective repulsive Cl⁻ intrinsic characteristic of CoP, the catalyst maintains good catalytic performance with saturated salt concentration, and the overpotential increasing is less than 28 mV at 10 mA cm⁻² from 0 M to saturated NaCl in electrolyte. Furthermore, the catalyst for seawater splitting performs superior corrosion-resistance with a low solubility of 0.04%. This work sheds fresh light into the development of efficient HER catalysts for salinity tolerance hydrogen evolution.

Hydrogen energy, as a renewable clean energy with high calorific value, has been regarded as one of the most ideal alternatives to replace traditional fossil fuel[1–3]. The most promising technique for producing hydrogen is electrochemical water splitting, which makes effective use of intermittent solar and wind energy[4–6]. Up to date, high-purity water is still used as the raw material in commercial electrocatalytic hydrogen production. However, the scarcity of freshwater resources and the high cost of purification process would restrict the commercial application of electrochemical splitting water technology[7].

Seawater electrolysis is regarded as a suitable replacement for other methods of producing hydrogen due to the fact that 97% of the world's water supply is found in the planet[8]. Furthermore, it is feasible to avoid the costly desalination of seawater that is necessary for present industrial water-splitting technology. However, due to the complicated chemical components of seawater, where large amounts of dissolved ionic salts, especially chloride ion, tend to poison catalysts and reduce the durability of seawater splitting[9]. Additionally, catalytic activity of catalysts decreases even more with the increase of salt concentration during the seawater splitting. Therefore, the development of highly effective and stable electrocatalysts that can ensure high salt-tolerance and robust corrosion-resistance is necessary for the commercial application of seawater electrolysis.

Up to date, Pt-based materials are one of the state-of-the-art electrocatalysts for hydrogen evolution reaction (HER), while the high costs and scarcity of these noble-metal catalysts prevent their industrial scale production[10,11]. Great efforts have been made to explore the cost-effective, highly active, and stable alternatives to Pt-based

[1]State Key Laboratory of Solidification Processing, Center of Advanced Lubrication and Seal Materials, School of Materials Science and Engineering, Northwestern Polytechnical University, Xi'an, Shaanxi 710072, P. R. China. [2]Institute of Material and Chemistry, Ganjiang Innovation Academy, Chinese Academy of Sciences, Ganzhou 341000, China. [3]These authors contributed equally: Xinwu Xu, Yang Lu. ✉e-mail: junqin.shi@nwpu.edu.cn; xlhuang@gia.cas.cn; heyibo@nwpu.edu.cn

catalysts for electrocatalytic seawater splitting, including alloys[12,13], composite materials[14,15], carbides[16–18], nitrides[19–21], oxides[22–24], phosphides[25,26], sulfides[27–29], and carbon-based materials[30,31], etc. Among these alternatives, transition metal phosphides (TMPs) with superior activity and long-term durability have attracted much attention and demonstrated as promising HER electrocatalysts[32,33]. In the family of TMPs materials, cobalt phosphide has attracted wide attention since these negatively charged phosphorus atoms effectively capture protons and promote the release of $H_2$[34]. These works effectively improved and maintained the stability and high activity of electrocatalysts in seawater, but most of them only studied the catalytic performance of catalysts in low salt concentrations. Unfortunately, in the actual process of seawater electrolysis, the concentration of saline ions in the electrolyte increases considerably as seawater electrolysis continues. As a result, the use of seawater still faces great issues since rising ions will enhance the competition between $H_2O$ molecules and active sites on the catalyst surface, obstructing or even poisoning the active site and drastically reducing the catalyst's electrocatalytic performance[35].

Herein, we found CoP can repel chloride ions while attracting $H_2O$ molecules to form a thin water layer on the catalyst surface based on molecular dynamics simulation. Following this discovery, a high-efficiency electrocatalyst was designed and constructed by a facile preparation strategy, which induced the CoP to grow homogeneously on the surface of rGO nanosheets. Meanwhile, the Ti fiber felt was introduced as substrate to support the CoP/rGO composites for constructing a binder-free CoP/rGO@Ti (CoPGT) electrode for seawater splitting. As expected, the as-prepared CoPGT catalyst with good structure stability and high catalytic activity required only a small overpotential of 103 mV at the current density of 10 mA cm$^{-2}$ and stable long-term performance in alkaline media. Even when the current density increased to as high as 200 mA cm$^{-2}$, the as-prepared catalyst can also realize a low overpotential of 210 mV, which is quite lower than that of the 20% Pt/C catalyst (300 mV). Importantly, attributed to the effectively repulsive effect of CoP toward Cl$^-$, the catalyst can maintain good catalytic performance and stability with salt concentration increasing, and the overpotential increase was <28 mV at 10 mA cm$^{-2}$ from 0 M to saturated chloride sodium in electrolyte. More surprisingly, it especially showed superior corrosion-resistance with a low solubility of 0.04% during the seawater splitting process compared to 20% Pt/C catalysts with the solubility of 2.37%, which is in agreement with the simulation results.

## Results and discussion

### Molecular dynamics simulation

Classical molecular dynamics simulation studies under simulated seawater conditions were conducted to predict the distribution of molecules and ions on the surface of CoP catalyst. As shown in Fig. 1a, salt ions are expelled to the outside of the water layer when the electric potential is −1.0 V Å$^{-1}$, while $H_2O$ molecules preferentially adsorb on the surface of the CoP catalyst in a region of several atoms, forming a thin water layer. The same adsorption layer of $H_2O$ molecules also exists in the other two cases of 0 and −0.1 V Å$^{-1}$ as verified by the density profile of $H_2O$ in Supplementary Fig. 1. The presence of the water layer without other ions can significantly improve HER efficacy and enhance catalyst corrosion-resistance because $H_2O$ molecules first adsorbs on catalysts, and then splitting into $H_2$ and OH$^-$ during the HER process in alkaline media[36]. The density profile of various ions (Fig. 1b) clearly displays that all these ions are separated by the water layer to move away from the CoP catalyst surface. Additionally, the anion ion repulsion effect becomes stronger as the electric potential is increased from 0 V Å$^{-1}$ to −1.0 V Å$^{-1}$. Because fewer ions are competing for on the catalyst surface as a consequence of this behavior, CoP catalyst is significantly more resistant to salt accumulation in electrolyte during

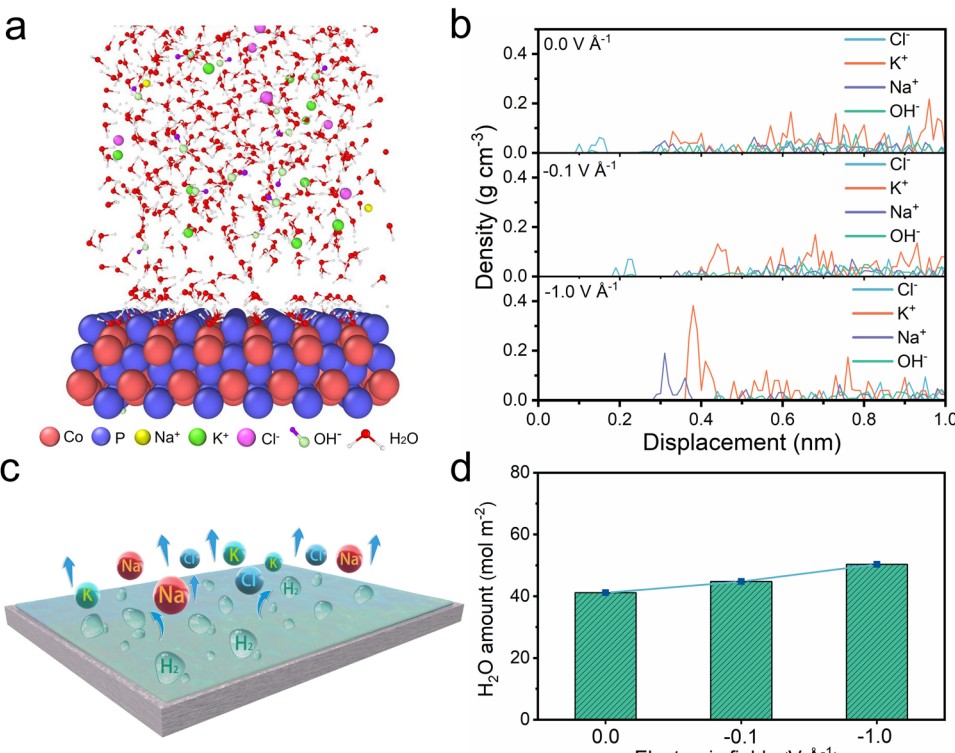

**Fig. 1 | MD simulations. a** Equilibrium configuration of electrolyte system (1.0 M KOH + 0.6 M NaCl) above the electrode surface of CoP with the presence of static external electric fields (−1.0 V Å$^{-1}$), viewed from XZ cross-section. **b** Mass density of various ions versus distance above the electrode surface with the presence of static external electric fields (0, −0.1, and −1.0 V Å$^{-1}$). **c** Schematic of salinity tolerance. **d** Amount of $H_2O$ within 5 Å above the electrode surface with the presence of static external electric fields (0, −0.1, and −1.0 V Å$^{-1}$).

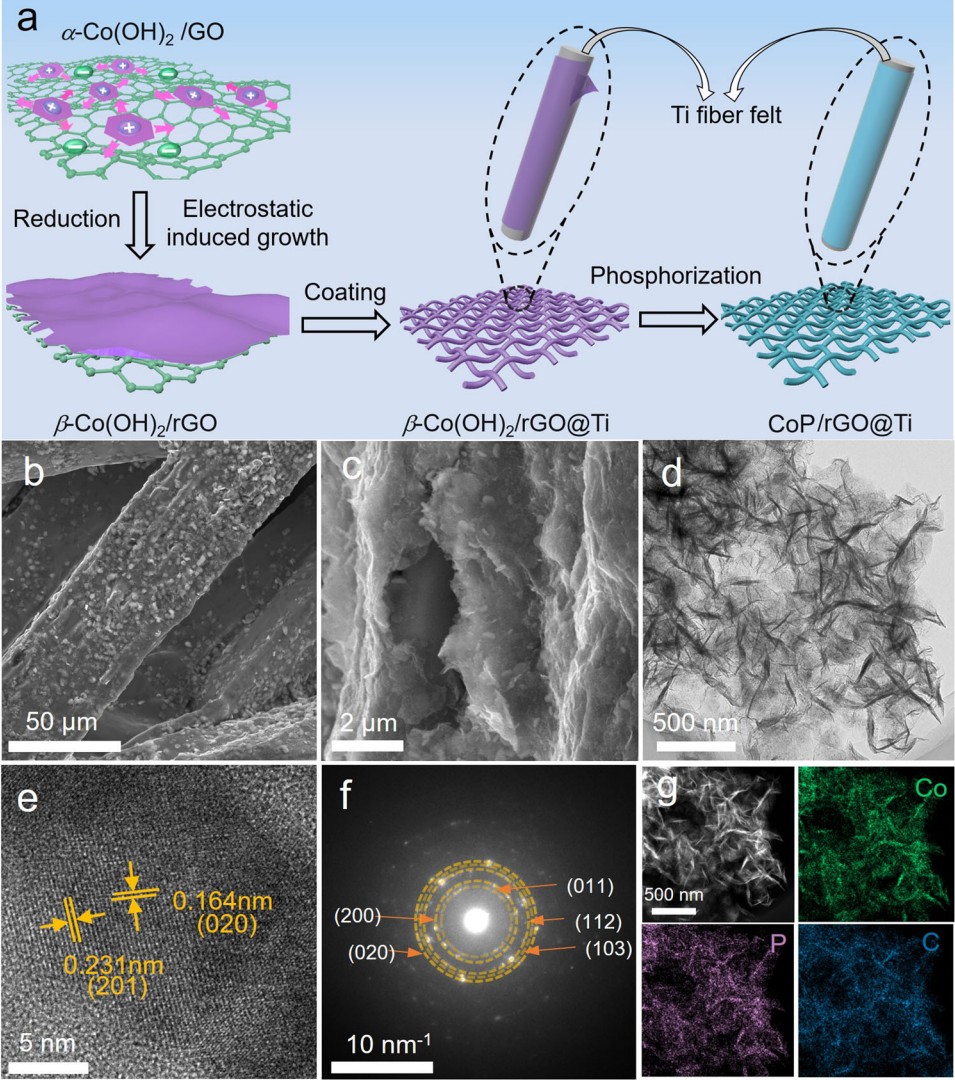

**Fig. 2 | Structural characterization of CoPGT. a** Schematic illustration of the synthesis process for CoPGT. **b-c** Low magnification and high magnification SEM images of CoPGT. **d–f** TEM image, HRTEM image and electron diffraction pattern of CoP/rGO. **g** Elemental mapping images of Co, P, and C elements in CoP/rGO.

the seawater splitting process. As displayed by the schematic diagram of salinity tolerance of CoP catalyst in Fig. 1c, the interference from salt ions is avoided in the process of water splitting, which contributes to the perfect electrocatalytic performance. Interestingly, the statistic amount of $H_2O$ molecules within 5 Å distance from CoP surface shows very slight enhanced with increasing electric potential, indicating the strong adsorption ability of CoP catalyst to $H_2O$ molecules (Fig. 1d). As a contrast, various ions on the surface of the $Co_3O_4$ catalyst are randomly distributed without forming an obvious water layer (Supplementary Fig. 2). A large number of ions adsorb on the surface of the catalyst and compete the active sites with $H_2O$ molecules on the surface, thus hindering the electrolytic water splitting reaction. The growing ions content on the surface of the $Co_3O_4$ catalyst with the increase of static external electric fields is not conducive to water splitting, suggesting that the $Co_3O_4$ catalyst cannot be resistant to salinity. Namely, it is vulnerable to the interference by salt ions, thus cannot guarantee the process of electrochemical splitting water.

## Preparation and characterization of CoPGT

According to the above finding based on the molecular dynamics calculations, a binder-free CoP/rGO@Ti (CoPGT) electrocatalyst was designed and fabricated wherein CoP uniformly dispersed on rGO

nanosheets wrapped around on the surface of Ti fiber felts. Figure 2a shows the detailed preparation process of CoPGT electrocatalyst. Firstly, the $β$-Co(OH)$_2$/rGO hybrid materials were prepared via an electrostatic-induced spread growth method by using the ultrathin α-Co(OH)$_2$ nanosheets and graphene oxide (GO) nanosheets as precursors (Supplementary Fig. 4 and Fig. 5). Then $β$-Co(OH)$_2$/rGO composite was wound on the Ti fiber felt fully and homogenously by a simple infiltration method. After phosphatizing, the CoPGT catalyst was obtained, which is becoming more compact compared with original clear and smooth Ti fiber surface (Supplementary Fig. 6). Figure 2b, c shows the scanning electron microscopy (SEM) images of CoPGT catalyst, it can be seen that the CoP/rGO nanoplates are densely covered on the Ti fiber. The corresponding elemental mapping images further reveal that Co, P, and C elements are uniformly distributed on the Ti fiber surface (Supplementary Fig. 7). The $Co_3O_4$GT and CoPT catalysts were prepared for comparison (Supplementary Figs. 8 and 9). Both of them present a similar frame structure with CoPGT. The corresponding elemental mapping images show the $Co_3O_4$GT with uniform distribution of Co, P, and C elements and the CoPT with uniform distribution of Co and P, which are beneficial to exert their catalytic performance. In order to obtain more structural information about CoP/rGO, transmission electron microscopy (TEM) characterization

was carried out. As shown in Fig. 2d, the CoP/rGO composite retains the sheet-like morphology, which is muti-layer and wrinkle with CoP coated uniformly and completely on the surface of graphene sheets. The high-resolution transmission electron microscopy (HRTEM) image of CoP/rGO presents two lattice fringes of 0.231 nm and 0.164 nm (Fig. 2e), agreeing well with d-spacing values of the (201) and (020) planes of CoP, respectively[37,38]. The selected-area electron diffraction (SAED) pattern further suggests the crystalline nature of the CoP. Figure 2f shows several bright diffraction rings from the inside to the outside can be indexed to the (011), (200), (112), (103), and (020) planes of CoP, respectively, indicating the polycrystalline characteristics of the CoP. In addition, the corresponding elemental mapping images of CoP/rGO clearly evidence that Co, P, C elements are well-dispersed (Fig. 2g). Co has a better dispersion on graphene, which is due to the electrostatic induced growth method to ensure that ultra-thin $\beta$-Co(OH)$_2$ nanosheets are tightly and uniformly coated on graphene, thus avoiding agglomeration.

The X-ray diffraction (XRD) pattern of CoP/rGO is shown in Fig. 3a. A series of diffraction peaks located at 31.7°, 36.5°, 46.3°, 48.4°, and 56.6° were observed, which are well indexed to the (011), (111), (112), (211), and (301) crystal planes of CoP phase (JCPDS Card No. 29-0497), further indicating that the β-Co(OH)$_2$ have been successfully converted to CoP. The diffraction peak marked with "*" near 40.6° is attributed to the (201) crystal face of Co$_2$P[32], indicating the presence of a small amount of Co$_2$P. Meanwhile, the pore structure of the sample was further tested by nitrogen adsorption-desorption experiment, the results are shown in Supplementary Fig. 10. It can be seen that the Brunauer-Emmett-Teller (BET) surface area of CoP/rGO is higher than double that of pure CoP powder, which is attributed to the avoiding agglomeration by the addition of rGO. The Barrett-Joyner-Halenda (BJH) pore-size distribution curve of CoP/rGO shows a higher pore volume, indicating a higher electrochemical surface area, which is beneficial to enhance the active surface area of catalyst.

We further investigated the chemical states of CoGPT using X-ray photoelectron spectroscopy (XPS). As shown in Supplementary Fig. 11, the Co, P, C, and O elements are detected from the XPS survey spectrum of CoPGT. Two pairs of peaks can be observed in the high-resolution Co 2p spectrum in Fig. 3b. The peaks at binding energies of 779.6 eV and 795.1 eV are assigned to Co with a partial positive charge for CoP[39]. As for the peaks at 781.8 eV and 797.6 eV, they are assigned to the oxidized Co species, which could be ascribed to surface oxidation due to the hydroxyl groups and absorbed water[40]. The XRD pattern and XPS spectra of Co$_3$O$_4$GT in Supplementary Fig. 12 confirm that the oxidized Co species is attributed to Co$_3$O$_4$. Besides, the peaks at 785.4 eV and 802.3 eV of Co satellite peaks are attributed to shake up excitation of the high-spin Co$^{2+}$ ions in the sample[41]. For the deconvoluted P 2p spectrum (Fig. 3c), the two peaks center at 129.9 eV and 130.8 eV corresponding to the P 2p$_{3/2}$ and P 2p$_{1/2}$ are well assigned to P with a partial negative charge of phosphides, while the peak located at 133.5 eV correspond to the phosphate due to surface oxidation[42,43]. In addition, the C 1s spectrum presents one peak around 284.8 eV, which is assigned to C-C from graphene (Fig. 3d). Meanwhile, the other peak at 287.2 eV is fitted to C−O, which may be incomplete reduction of GO or partial oxidation of rGO[44].

## Electrochemical properties toward HER

The electrocatalytic performance of CoPGT as a working electrode was evaluated in 1.0 M KOH electrolyte. Figure 4a shows the polarization curves of various catalysts for alkaline HER. When the current density reached 10 mA cm$^{-2}$, the CoPGT shows a low overpotential of 103 mV, which is quite lower than that of Co$_3$O$_4$GT (260 mV), CoPT (140 mV) and CoP catalyst ink (203 mV). The better catalytic performance of CoPGT than that of CoPT and CoP is mainly attributed to the large specific surface area and good electrical conductivity provided by

graphene (Supplementary Fig. 13). The bare Ti fiber felt and rGO powder exhibit poor electrocatalytic activity, which is almost negligible. Interestingly, the electrocatalytic performance of CoPGT increases with the increase in overpotential faster than that of 20% Pt/C, indicating that the HER performance of CoPGT is superior to that of 20% Pt/C at a high current density. The Tafel plots of various catalysts were investigated to further understand the HER mechanism (Fig. 4b). It can be seen that the CoPGT exhibits a Tafel slopes of 66.1 mV dec$^{-1}$, which is comparable to that of CoPT (63.1 mV dec$^{-1}$) and CoP (67.1 mV dec$^{-1}$). Additionally, the Tafel slope reveals that the HER process of CoPGT is through a Volmer-Heyrovsky mechanism, in which the rate-determining step on electrodes is an electrochemical reaction with an adsorbed hydrogen atom and a proton to create H$_2$[45]. The large Tafel slope of the Co$_3$O$_4$GT (177.4 mV dec$^{-1}$) is resulted from the low catalytic activity of Co$_3$O$_4$. The obtained Tafel slopes of 20% Pt/C is about 37.8 mV dec$^{-1}$, which is consistent with the reported values[46]. In addition, as shown in Supplementary Fig. 14, the faradic efficiency of CoPGT catalyst for HER is estimated to be close to 100%, indicating that almost all electrons are utilized for producing hydrogen.

The cyclic stability of CoPGT was evaluated by comparing the LSV curves before and after 10000 cycles in 1.0 M KOH. As shown in Fig. 4c, the polarization curve of the CoPGT catalyst shows no significant change and the overpotential is increased by only 10 mV at the current density of 10 mV cm$^{-2}$, indicating its stable catalytic performance. To further assess the long-term durability of CoPGT, chronopotentiometric (CP) tests at a fixed current density of 10 mA cm$^{-2}$ were performed. Note that the catalyst performs efficiently without notable loss during long-term electrolysis (Fig. 4d), indicating the improved stability of CoPGT electrocatalyst. Especially, even the current density increased to 200 mA cm$^{-2}$, the performance of CoPGT still shows no obvious degradation, which is far more stable than that of 20% Pt/C. Furthermore, there are no apparent change in morphology and surface chemical state of the post-HER CoPGT, as revealed by the SEM image (Supplementary Fig. 15), the TEM image (Supplementary Fig. 16), and XPS spectra (Supplementary Fig. 17). All above results demonstrate that CoPGT is a highly active and stable electrocatalyst for alkaline HER.

## Salinity resistance of the catalysts

The electrocatalytic performance of CoPGT electrocatalyst in the electrolyte with chloride sodium (NaCl) concentration from 0 M to saturation was investigated. As expected, with the increase of NaCl concentration in the electrolyte, CoPGT and CoPT show more stable catalytic performance compared with Co$_3$O$_4$GT (Fig. 5a–c). The increase of overpotential for CoPGT and CoPT have a slight decline of about 28 mV and 38 mV from 0 M to saturated NaCl in electrolyte, respectively, which are far less than that of Co$_3$O$_4$GT (80 mV) as shown in Fig. 5d, e. The CoPGT catalyst has a remarkable resistance ability to salinity along with the increase of salt concentration, which is highly in agreement with the simulation results. Conversely, Co$_3$O$_4$GT catalyst is easily disturbed by saline ions, and displays a rapid performance degradation. In order to further demonstrate the salinity tolerance of CoP, the electrocatalytic properties of CoP catalyst ink was also studied in the electrolyte with chloride sodium (NaCl) concentration from 0 M to saturation. As shown in Supplementary Fig. 18, the catalytic performance of CoP catalyst ink has no obvious attenuation with increasing concentration of NaCl as expected. The TOF of CoP/rGO is 0.0239 s$^{-1}$, indicating a high intrinsic activity. To figure out the origin of the enhanced activity of CoP/rGO electrocatalyst, the double-layer capacitances (C$_{dl}$) measurements were carried out to evaluate the electrochemical active surface areas (Supplementary Fig. 19)[47]. The CoP/rGO had a higher C$_{dl}$ of 2.33 mF cm$^{-2}$ than CoP, indicating more electroactive surface exposed in CoP/rGO compare with pure CoP. The electrochemical impedance spectroscopy (EIS) was performed to deeply study the charge-transfer mechanism and the

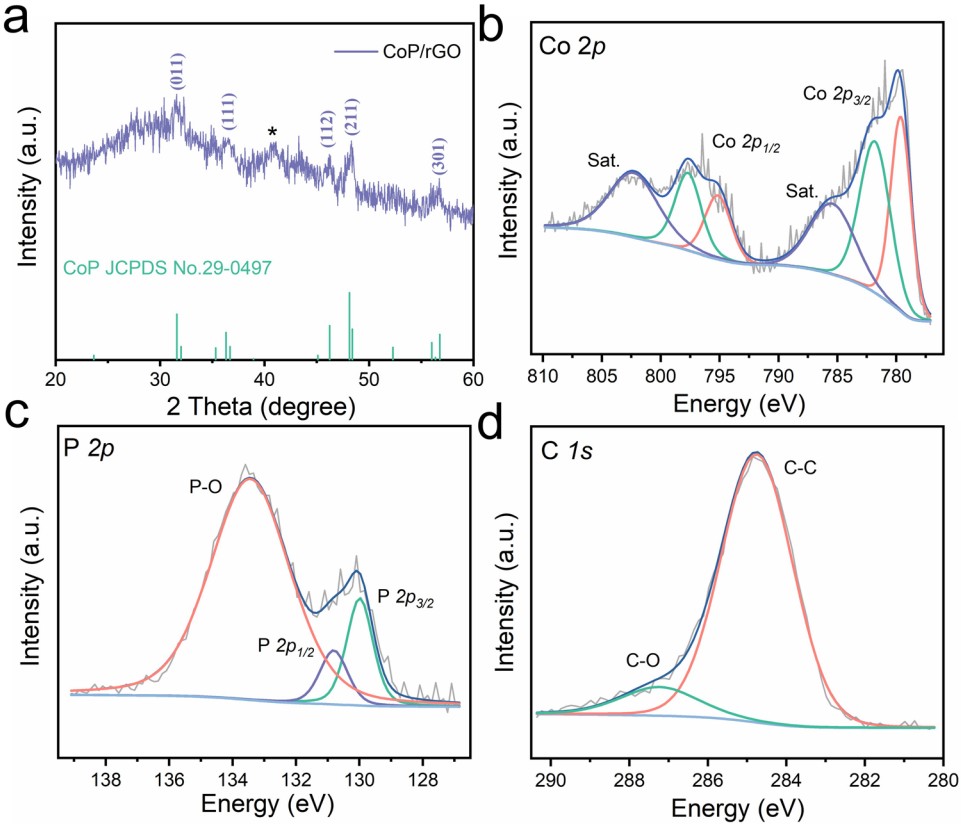

Fig. 3 | **Structural characterization of CoPGT. a** XRD pattern of CoP/rGO. **b**–**d** High-resolution Co *2p*, P *2p*, and C *1s* XPS spectra of CoPGT.

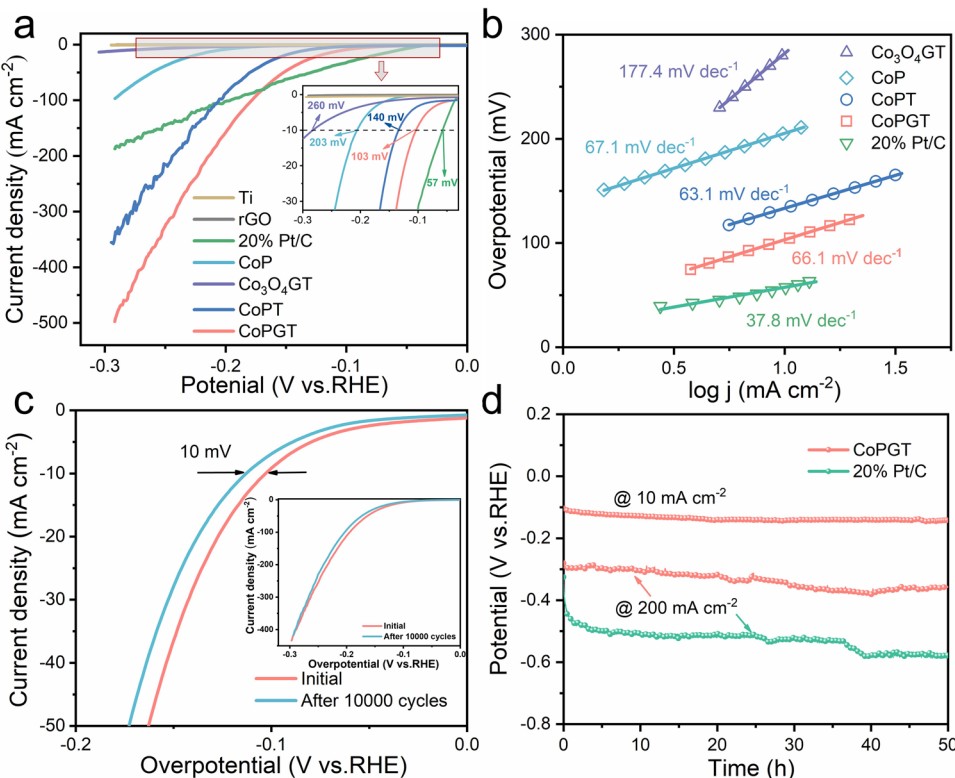

Fig. 4 | **Electrochemical HER performance measurements. a** LSV curves of CoPGT, CoPT, CoP, Co₃O₄GT, rGO, Ti and 20% Pt/C for the HER in 1.0 M KOH, inset: the detailed at −10 mA cm⁻². **b** The corresponding Tafel plots. **c** Polarization curves of CoPGT initially and after 10,000 cycles at a scan rate of 1 mV s⁻¹ for HER, inset: overall view of LSV curves. **d** Chronopotentiometry curves of CoPGT and 20% Pt/C in 1.0 M KOH.

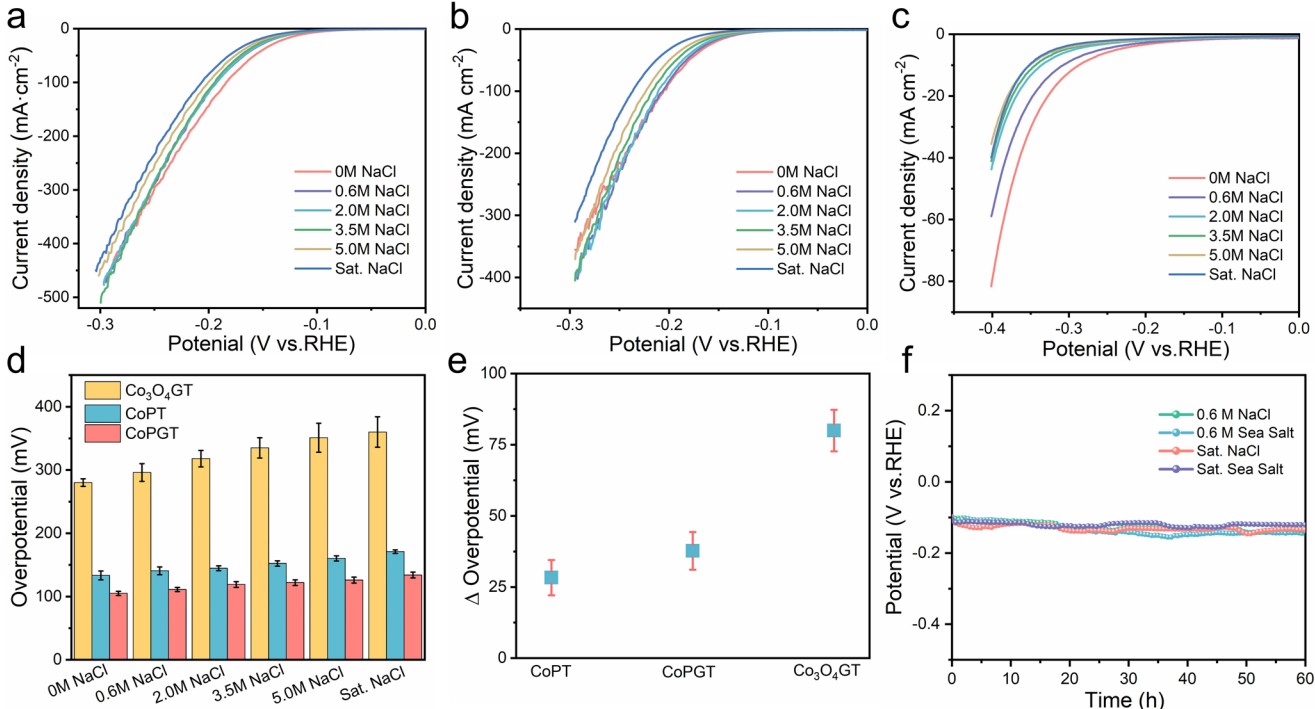

**Fig. 5 | Salinity resistance performance. a–c** LSV curves of CoPGT, CoPT, $Co_3O_4GT$ for the HER in different salt solutions, respectively. **d** Corresponding variogram of overpotential at 10 mA cm$^{-2}$. **e** Overpotential difference against salt concentration gradient for CoPGT, CoPT, $Co_3O_4GT$. **f** Chronopotentiometry curves of CoPGT at 10 mA cm$^{-2}$ in 1.0 M KOH with 0.6 M NaCl, saturated NaCl, 0.6 M sea salt and saturated sea salt, respectively.

resulting Nyquist plots were shown in Supplementary Fig. 20. Apparently, the CoP/rGO catalyst presented a smaller $R_{ct}$ value than CoP, implying faster charge transfer. In addition, we also explored the stability of the CoPGT electrocatalyst under different salt concentrations. As shown in Fig. 5f, the catalyst still performed well without significant loss in simulated and real seawater during the 60 h CP test, which is much better than 20% Pt/C (Supplementary Fig. 22), indicating the superior structure stability of CoPGT. Meanwhile, the CoPGT electrocatalyst exhibits a negligible overpotential variation after continuous LSV scanning at 10 mA cm$^{-2}$ no matter in NaCl solutions or sea salt solutions (Supplementary Figs. 23 and Fig. 24), further demonstrating its good salinity resistance. In contrast, the overpotentials of the 20% Pt/C catalyst increased significantly after 10000 LSV cycles in 1.0 M KOH containing different concentrations of sea salt and NaCl (Supplementary Fig 25), suggesting its poor stability in harsh seawater environments.

The poor electrolytic performance stability of commercial 20% Pt/C in simulated seawater is mainly caused by the dissolution of Pt due to the aggressive corrosion of saline ions, thus losing part of the active site, which would face the stability degradation under long-term operation[48]. In view of this issue, we measured the concentration of platinum and cobalt elements in the electrolyte by an inductively coupled plasma mass spectrometry (ICP-MS) analysis after the 10 h CP test. As shown in Supplementary Fig. 21, the dissolution rate of Pt element reach up to 2.37%, while the dissolution rate of Co element in the electrolyte with 0.6 M NaCl is only about 0.04%, which is really quite negligible.

All the above results show that the CoPGT electrocatalyst has good corrosion-resistance and salt-tolerance, forcefully demonstrating that it is a good electrocatalyst candidate for seawater splitting. It should be pointed out that the CoPGT electrocatalyst compares favorably to many reported HER catalysts in alkaline seawater. The detailed performance comparison of CoPGT with the state-of-the-art HER electrocatalysts for hydrogen evolution is summarized in Supplementary Table 2.

In summary, CoP was found that can repel chloride ions while attracting $H_2O$ molecules to form a thin water layer on catalyst surface based on molecular dynamics simulation. Thus, a binder-free CoPGT electrode was designed and fabricated by a facile strategy, which induced the $\beta$-Co(OH)$_2$ uniformly grow on the rGO nanosheets via electrostatic induction and then turned into CoP/rGO wrapped around the surface of Ti fiber felt through phosphatizing. Attributed to the effectively repulsive effect of CoP towards Cl$^-$, the as-obtained CoPGT electrocatalyst exhibited good HER performance with a small overpotential of 210 mV at 200 mA cm$^{-2}$ and superior stability in alkaline electrolyte. Especially, it is remarkable that the overpotential increase of the catalyst is less than 28 mV with a rise in salt concentration from 0 M to saturated NaCl in the electrolyte, which is in accordance with the molecular dynamics simulation. Meanwhile, CoPGT electrode demonstrated improved corrosion-resistance with a low solubility of 0.04% compared to commercial 20% Pt/C catalysts with the solubility of 2.37% during the seawater splitting process. This work not only provides an appealing strategy to design an affordable and functional catalytic electrode, but also offers a fresh viewpoint on how to design electrocatalysts with good salt-tolerance and stability for seawater splitting.

## Methods

### Chemicals
Ammonia water (25%), graphene oxide (10 mg·mL$^{-1}$), cobaltous nitrate hexahydrate (Co(NO$_3$)$_2$·6H$_2$O), and hydrazine monohydrate (N$_2$H$_4$·H$_2$O, 85%) were supplied by Sinopharm Chemical Reagent Co., Ltd. (China). Potassium hydroxide (KOH, Aladdin, 99.99%), Pt/C (20 wt% Pt), sodium hypophosphite (NaH$_2$PO$_2$), Nafion (5 wt%), Sodium chloride (NaCl) and ethanol (C$_2$H$_5$OH) were purchased from Shanghai Aladdin Bio-Chem Technology Co., Ltd. (China). Titanium foil obtained from CeTech Co., Ltd. (China). All reagents were used as received without further purification. The water was purified through a Millipore system throughout all experiments.

## Preparation of α-Co(OH)$_2$

Ammonia water (2 mL, 25%) was added to a solution of Co(NO$_3$)$_2$ (240 mL, 0.05 M), the mixture was stirred for 5 min and then silence. Afterwards, the as-obtained green blue precipitate was washed with deionized water by centrifugation for 3 times, and then transferred into a 100 mL Teflon-lined autoclave, which was filled with solvents (water/methanol, 32 mL/32 mL). And then, these systems were sealed and heated at 180 °C for 30 min to obtain colloidal solution of α-Co(OH)$_2$ that can be used directly.

## Preparation of β-Co(OH)$_2$/Graphene hybrid materials

The β-Co(OH)$_2$/rGO hybrid was prepared using a electrostatic induced spread growth method[49], wherein homogenous and full coating of β-Co(OH)$_2$ nanosheets on graphene. Briefly, graphene oxide solution (10 mL, 9.6 mg of graphene oxide) and colloidal solution of α-Co(OH)$_2$ (100 mL, 96 mg of α-Co(OH)$_2$) were mixed by ultrasonic for 30 min, and then added hydrazine monohydrate (0.32 mL), subsequently refluxed at 95 °C for 24 h. Afterwards, the resulting β-Co(OH)$_2$/rGO hybrid was collected by centrifugation, washed several times with DI water and dispersed in DMF to the concentration of 3 mg mL$^{-1}$.

The preparation process for β-Co(OH)$_2$ was similar to that for β-Co(OH)$_2$/rGO except addition of graphene oxide, afterwards collected by centrifugation, washed 3 times with DI water, and then freeze-dry, followed by phosphorization to obtained CoP powder.

## Preparation of CoPGT electrode

Firstly, the Ti fiber felt (TF) was thoroughly soaked in the above solution of β-Co(OH)$_2$/rGO hybrid to let the active coat the surface, followed by drying in the oven. Repeat this process multiple times. To further prepare CoP/rGO@TF, the precursor β-Co(OH)$_2$/rGO@TF and 500 mg of NaH$_2$PO$_2$ were placed at two separate positions in a porcelain with NaH$_2$PO$_2$ at the upstream side and downstream side of the tube furnace, respectively. The boat was put into a tube furnace, followed by phosphorization at 350 °C for 2 h at a heating rate of 10 °C min$^{-1}$ under the flow of N$_2$. Finally, the furnace was naturally cooled down to room temperature. The resultant CoP/rGO@TF was rinsed sequentially with DI water, ethanol and then dried in a vacuum oven. The average CoP/rGO loading mass was measured to be 1.5 mg cm$^{-2}$.

For comparison, CoP@TF, Co$_3$O$_4$/rGO@TF materials were also prepared with the same method without the addition of graphene oxide, NaH$_2$PO$_2$, respectively. The average CoP, Co$_3$O$_4$/rGO loading mass was measured to be 1.5 mg cm$^{-2}$, 1.25 mg cm$^{-2}$, respectively. 3 mg of 20% Pt/C electrocatalyst was dispersing in a mixture solution containing 16 uL of 5 wt% Nafion solution and 1 mL of DI water. The mixture was then ultrasonicated for 30 min to generate a homogenous ink, and then 10 uL of uniform catalyst inks were coated on a GCE and dried in oven. For the sake of writing, the as-prepared CoP/rGO@TF, Co$_3$O$_4$/rGO@TF, and CoP@TF were abbreviated as CoPGT, Co$_3$O$_4$GT, and CoPT, respectively.

## Characterizations

The crystal phases of the materials were characterized by XRD tests obtained on D8 Advance Power diffractometer (BRUKER, German, using Cu-Kα radiation, $\lambda = 1.5418$ Å). The morphology and chemical element composition of the materials were observed by scanning electron microscopy (SEM) and energy dispersive X-ray spectrometry (EDS), respectively, which were performed on FEI Helios G4 CX Scanning electron microscope (FEI, USA). The nanoscale crystal structure of the materials was obtained using a transmission electron microscopy (TEM) equipped with EDS (Talos F200X, Thermo Fisher). The chemical binding states of various ions were analyzed by X-ray photoelectron spectroscopy (XPS), which was employed on a PHI 5000 versa Probe III spectrometer (ULVAC-PHI, Japan). The Brunauer-Emmett-Teller (BET) specific surface areas of the materials were measured on (Micromeritics ASAP 2020) by nitrogen adsorption at 77.4 K. The samples

were degassed for 3 h at 300 °C prior to measurements. The inductively coupled plasma mass spectrometry (ICP-MS) measurement was conducted on USA-Aglient-7800(MS). The conductivity is measured with a four-probe tester and then obtained according to equation: $\sigma = 1/(R \cdot S)$, in which CoP/rGO powder and CoP powder are pressed into thin slices.

## Electrochemical measurements

All the electrochemical measurements were performed on a CHI660E electrochemical workstation (Chenhua Equipment Co., China) by using a three- electrode electrochemical cell. The as-prepared catalysts, graphite paper, and Hg/HgO electrode were used as the working electrode, auxiliary electrode, and reference electrode, respectively. The electrolyte is 40 mL 1.0 M KOH with pH≈14. All potentials were converted into reversible hydrogen electrode (RHE), using the following equation: $E$ (RHE) = $E$ (Hg/HgO) + 0.098 + 0.059 × PH. The linear sweep voltammetry (LSV) curves were obtained with a sweep rate of 1 mV s$^{-1}$. Electrochemical impedance spectroscopy (EIS) was measured at a frequency range from 0.1 Hz and 10$^6$ Hz. All the polarization curves were recorded with 85% iR compensation using the solution resistance estimated from EIS results.

The stability of CoPGT and 20% Pt/C was tested by repeating LSV running at a scan rate of 100 mV s$^{-1}$ with the potential range between -0.80 and -1.20 V (vs RHE). The electrochemical double-layer capacitance ($C_{dl}$) of catalysts were evaluated by using cyclic voltammetry in a non-Faradaic region from −0.40 to −0.50 V vs. Hg/HgO at different scan rates ranging from 10 to 60 mV s$^{-1}$. The $C_{dl}$ equals the resulting linear slope of the difference between the half of the current density of anodic and cathodic at −0.45 V vs. Hg/HgO versus scan rate. The turnover frequency (TOF) value is estimated by the following equation: TOF = $\frac{j \times S}{Z \times N \times F}$. Here, $j$ is the current density, $S$ is the area of the electrode, $Z$ represents the electrons transfer number in HER which is 2, $F$ is the Faraday constant (96485.3 C mol$^{-1}$), and $N$ is the number of sites of the active materials. The content of Co element was 21.82 wt%, as measured by inductively coupled plasma mass spectrometry (ICP-MS).

The Faraday efficiency (ε) of HER was determined by measuring the amount of H$_2$ produced. The Faraday efficiency (FE) can be calculated as follows: $FE_{H_2} = \frac{n_{H_2} \times Z \times F}{Q_s}$. Where $n_{H_2}$ is the amount of the hydrogen, which was measured using gas chromatography (8860 GC). The catalytic gas is collected for 90 min and measuring the amount of H$_2$ produced every 10 min. $Z$ is the number of transferred electrons ($Z = 2$ in this work); $F$ is the Faraday constant, $Q_s$ is the total charge in the reaction for 60 min, which was calculated by the integration of Chronoamperometry curve. The Faradaic efficiency tests were performed in a two-electrode system using an electrochemical workstation (CHI 660E). The CoPGT catalyst and Pt foil were used as the counter electrode (CE) and reference electrode (RE), respectively.

The electrode durability was tested by the galvanostatic method, which was conducted at 10 mA cm$^{-2}$ for 60 hours. The tests were performed in 1.0 M KOH solution. All the electrochemical measurements were carried out at room temperature.

## Details of the theoretical simulations

Molecular dynamic (MD) simulations were performed to study the ion distributions on the catalyst surface of CoP and Co$_3$O$_4$. The periodic boundary conditions were applied in the horizontal plane. A few layers of CoP or Co$_3$O$_4$ were placed at the bottom of the box and fixed as the model of electrode surface. To simulate the electrolyte in the experiment (1 M KOH + 0.6 M NaCl), the solvent model contains 26352 H$_2$O molecule, 251 Na$^+$, 251 Cl$^-$, 479 K$^+$ and 479 OH$^-$. The size of model box containing CoP is about 131.96 Å × 122.73 Å × 127.50 Å, and the one containing Co$_3$O$_4$ is about 131.34 Å × 123.26 Å × 165.20 Å. In all simulations, the system was run at 300 K with NVT ensemble for 3 ns to reach equilibrium. A variable electric field in Z direction was applied to each atom with selected voltage values of 0, 0.1 and 1 V Å$^{-1}$, respectively. The

interatomic interactions of catalyst with $H_2O$ and various ions were described by Coulomb and Lennard-Jones potentials, and the corresponding parameters of $Na^+$、$Cl^-$、$K^+$ were developed by In Suk Joung and Thomas E Cheatham[50], as listed in Supplementary Table 1. The interaction between $H_2O$ molecules was described by SPC/E model[51]. The interaction within $OH^-$ are derived from the CVFF force field and the interaction parameters of CoP and $Co_3O_4$ are referred according to previous report[52,53], all above details are shown in the Supplementary Table 1. In addition, the detailed simulation models of both CoP and $Co_3O_4$ catalysts are shown in Supplementary Fig. 3. The cutoff distance was 10 Å for both two potentials, and time step was 1 fs. All the MD calculations were implemented in large-scale atomic/molecular massively parallel simulator (LAMMPS)[54], and the configurations were visualized by OVITO software[55].

## Data availability

The data supporting the findings of this study are available from the corresponding authors upon reasonable request.

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

## Acknowledgements

We sincerely appreciate the financial support from the Fundamental Research Funds for the Central Universities (D5000210129) and State Key Laboratory of Solidification Processing. Y.H. acknowledges the National Nature Science Foundation of China (22109127) and the Chinese Postdoctoral Science Foundation (2021M702666). Y.H. is also very grateful for the support from the Youth Project of "Shaanxi High-level Talents Introduction Plan".

## Author contributions

X.X., X.H., and Y.H. conceived the project and designed the experi-ments. X.X. performed the experiments, analyzed the data and wrote the manuscript. Y.L. and J.S. carried out the MD simulation calculations. Z.M. and X.H. carried out the microscopy analysis. K.Y. and C.L. assisted to materials preparation. T.Z. and D.Z. contributed to the data analysis. J.S., X.H. and Y.H. jointly supervised the work and revised the manuscript.

## Competing interests

The authors declare no competing interests.
