## [Peer review file · Nature Communications]

REVIEWER COMMENTS

Reviewer #1 (Remarks to the Author):

This work developed a salinity-tolerant electrocatalyst by uniformly growing CoP on rGO nanosheets supported by Ti fiber felt. Molecular dynamics simulations revealed that CoP repels chloride ions while attracting H₂O molecules, resulting in the formation of a thin water layer on the catalyst surface. This phenomenon enhances the HER performance and stability of the catalyst in seawater. However, the materials designing strategy is not new and the seawater splitting is not economically meaningful. Some main issues are listed in the following.

1. Fig. 3b has a problem with the red peaks not meeting the desired peak area ratio between 2p_{1/2} and 2p_{3/2}. The same problem is observed in Fig. 3c.
2. In Fig.4, the sample of CoPGT exhibits superior HER performance compared to CoPT, but its Tafel slope is higher. Why is this the case?
3. There is no indication in Fig. 5f which line is Pt/C and which line is sample.
4. The XRD peaks in Fig. 3a are too weak, and some peaks, like the one around 40°, do not match to a standard card. The author should explain those peaks.
5. In the stability test depicted in Figure 4d, the authors should explain why the potential of Pt/C decreased between 6 to 8 hours.

Reviewer #3 (Remarks to the Author):

The paper is an exciting topic, which has some interesting findings, and given the interest and importance of cobalt phosphides in water splitting, I am sympathetic with the view to publish. Focusing on the molecular dynamics (MD) component. The result that the surface of CoP excludes the halide is noteworthy and interesting. However, I consider that there are 2 major problems with the manuscript/SI that need addressing before the paper is accepted for publication with the MD included. Indeed, as a consequence of these, I do not find the MD results sufficiently convincing. The first issue is that there is insufficient information to be able to reproduce the findings. The major omissions include a proper description of the force field and the surfaces. I appreciate the salt-water parameter set is referenced, but not the CoP and Co₃O₄ interaction parameters with themselves and critically, with water, chloride and alkali ions. What are the parameters used, what is the evidence that they are sensible? This is particularly important given that the metal phosphides are generally metallic. I presume, as "A few layers of CoP or Co₃O₄ were placed at the bottom of the box and fixed as the model of electrode surface", that the coordinates of the cobalt compounds were held fixed, if not then I would expect to see interaction parameters between all of the species. The other important piece of information missing is what surfaces were chosen? The ordering or otherwise of water on the surfaces (CoP and Co₃O₄) will depend on which surface structure, and arbitrarily selecting a disordered spinel surface for Co₃O₄ and a highly ordered CoP may increase the chance of getting the desired result. Also, why are those particular surfaces chosen? Is there evidence that they are sensible surfaces? This becomes an even more critical assumption when the surface atoms held fixed. In addition to answering the questions raised above, even if in the SI, please include input files for the 2 simulation cells, as this would also give the precise parameters chosen and help other researchers reproduce and develop the work. The second area of concern is the resulting water structure next to the CoP. However, I appreciate that I am inferring what I see from the projections. But it looks like the water is over bound, due to a poor representation of the electrostatics and short range components leading to unphysical water ordering that then results in exclusion of the salt ions. Including rdfs for the Co and P with the oxygen and hydrogen of water would resolve this, so too the parameters for the model. Finally, I was confused about the periodicity used, was it 2D or 3D? Particularly, if 3D, rather than just stating that the field is -1 V/Å, on your structural figure, identify which is the negative and positive sides.

Reviewer #4 (Remarks to the Author):

In this study, the authors provide a new CoP/rGO@Ti electrode for the hydrogen evolution reaction in seawater electrolysis. The HER activity of CoP/rGO@Ti electrode in alkaline seawater electrolyte is excellent due to the highly effective repulsive Cl⁻ intrinsic characteristic of CoP. The results are interesting and may contribute well to the electrocatalytic community in producing hydrogen using the seawater-based electrolyte. However, there are some issues that should be made clear before publication in Nat.Com as follows.

1. Please add more detailed information about the materials used in the experiment. In addition, what is the purity of KOH?
2. For comparison of the HER activity of the prepared catalyst with the 20wt% Pt/C, the 20wt% Pt/C should be deposited on TF (1.5mg/cm²) to record its HER activity. I believe that the low loading of Pt/C tested in the RDE system leads to low HER performance at a high current density.
3. The morphology of Co₃O₄GT and CoPT catalysts should be proved by SEM images to compare with that of CoPGT.
4. Please insert the enlarged Figure 4a to see the values of potential at a current density of -10 mA.cm⁻², which values were mentioned in the manuscript.
5. The time for the stability test is 10h to be short and not easy to say the stability of catalysts.
6. What is the potential range for cyclic stability? It is an important parameter for readers.
7. SEM image of CoPGT was shown in Supplementary Fig. 11 to see the unchanged morphology of CoPGT after the stability test. It would be better to show the TEM image of CoPGT for clarifying CoPGT morphology.
8. Why the values of overpotential difference against salt concentration gradient for CoPGT and CoPT are different as seen in Fig. 5 e, even though they have the same mechanism with the highly effective repulsive Cl⁻ intrinsic characteristic of CoP? Please the authors give comments.
9. Supplementary Fig. 9 and 15 are not mentioned in the manuscript.
10. Since values of electrochemically active surface area (ECSA), charge transfer resistance (R_{ct}), and turnover frequency (TOF) are very important parameters to evaluate the catalytic activity of HER catalysts. Please provide them.
11. To exclude the possibility of side reactions during the HER under half-cell measurement, the full cell test should be investigated for CoPGT. H₂ product should be collected and calculated the Faradaic efficiency during the full-cell operation of water splitting.

First of all, we sincerely appreciate all of these valuable comments and suggestions from the reviewers, which would undoubtedly help us to further improve the quality of our current manuscript.

Following are our responses and detailed explanations towards these comments from the reviewers.

Responses to Reviewers:

To Reviewer #1:	1-6
To Reviewer #3:	7-15
To Reviewer #4:	16-27
List of Revisions	28-30

Response to Reviewer#1:

General Comment: This work developed a salinity-tolerant electrocatalyst by uniformly growing CoP on rGO nanosheets supported by Ti fiber felt. Molecular dynamics simulations revealed that CoP repels chloride ions while attracting H₂O molecules, resulting in the formation of a thin water layer on the catalyst surface. This phenomenon enhances the HER performance and stability of the catalyst in seawater. However, the materials designing strategy is not new and the seawater splitting is not economically meaningful.

Author Reply: First of all, we sincerely appreciate all your professional and valuable comments and suggestions, which would undoubtedly help us to further improve the quality of our current manuscript. According to your kind and precious advices, corresponding revisions have been made in the manuscript. We sincerely hope that the revisions can meet your requirements and make the revised manuscript more suitable for publication in *Nature Communications*. Additionally, before answering the specific comments, we think that we should make an explanation on the originality and novelty of this work, since we believe that they are essentially important.

Actually, the intriguing innovation of this work is that we found the CoP can repel chloride ions while attracting H₂O molecules, resulting in the formation of a thin water layer on the catalyst surface. As you mentioned, this phenomenon enhances the HER performance and stability of the catalyst in seawater. Based on this discovery, we developed a salinity-tolerant electrocatalyst by uniformly growing CoP on rGO nanosheets supported by Ti fiber felt. Although the preparation of β -Co(OH)₂/rGO hybrid material referenced from our previous work [*Adv. Funct. Mater.* **23**, 4345–4353 (2013)], the construction of the binder-free CoP/rGO@Ti (CoPGT) electrode is

provided by this work. And the CoPGT exhibited excellent HER activity and outstanding stability in alkaline seawater.

As for the economic value of seawater splitting, it is indeed a heavily studied topic. For example, some analyses suggest that direct seawater electrolysis is not economic favorable [*Energy Environ. Sci.* **14**, 4831–4839 (2021); *ACS Sustain. Chem. Eng.* **7**, 8006–8022 (2019)], while some other studies suggest that direct seawater electrolysis shows low cost with economic benefit [*Nat Energy* **8**, 264–272 (2023); *Nat Commun.* **14**, 3607 (2023); *Nature* **612**, 673–678 (2022); *Nat. Energy* **5**, 367–377 (2020)]. At this stage, the community has not reached a convincing or fixed conclusion about the preferable method, and more studies are needed on this topic. Nevertheless, in terms of the richness of resources, seawater electrolysis is worthy to study and can be regarded as a suitable replacement for other methods of sustainable hydrogen gas (H₂) production, because seawater is one of the most abundant natural resources on our planet and accounts for 96.5% of the water on the earth.

Thus, we believe that the findings and results of this work may contribute well to the electrocatalytic community in producing hydrogen using the seawater-based electrolyte, which will arouse the remarkable enthusiasm of readers.

Specific responses and explanation are shown below:

Comment 1-1: Fig. 3b has a problem with the red peaks not meeting the desired peak area ratio between $2p_{1/2}$ and $2p_{3/2}$. The same problem is observed in Fig.3c.

Author Reply: Thank you very much for pointing out this problem. Fig. 3b and 3c have been redrawn and the corresponding peak positions have been corrected in the revised manuscript as follows:

Fig. 3 High-resolution b) Co 2p and c) P 2p XPS spectra of CoPGT.

“The peaks at binding energies of 779.6 eV and 795.1 eV are assigned to Co with a partial positive charge for CoP³⁹. As for the peaks at 781.8 eV and 797.6 eV, they are assigned to the oxidized Co species, which could be ascribed to surface oxidation due to the hydroxyl groups and absorbed water⁴⁰. The XRD pattern and XPS spectra of Co₃O₄GT in the Supplementary Fig. 12 confirm the oxidized Co species could be attributed to Co₃O₄. Besides, the peaks at 785.4 eV and 802.3 eV of Co satellite peaks are attributed to shake up excitation of the high-spin Co²⁺ ions in the sample⁴¹. For the deconvoluted P 2p spectrum (Fig. 3c), the two peaks center at 129.9 eV and 130.8 eV corresponding to the P 2p_{3/2} and P 2p_{1/2} are well assigned to P with a partial negative charge of phosphides, while the peak located at 133.5 eV correspond to the phosphate due to surface oxidation^{42,43}.”

Comment 1-2: In Fig.4, the sample of CoPGT exhibits superior HER performance compared to CoPT, but its Tafel slope is higher. Why is this the case?

Author Reply: Thank you for this comment. As is well known, the Tafel slope is a fitted value that depends on the selected range of current densities to some extent. In this work, the Tafel curves in Fig. 4b is derived from the LSV curves

in Fig. 4a. It can be seen that the HER performance of the two catalysts at low current density is very close, so their Tafel slope is also similar to each other. With the increase of current density, the advantage of CoPGT in HER performance gradually appears. When a wider range of current density is selected for fitting (Fig. R1), we can see that the Tafel slope of CoPGT (61.8 mV dec^{-1}) is much smaller than that of CoPT (75.6 mV dec^{-1}).

Fig. R1 Tafel plots of CoPGT and CoPT catalysts.

Comment 1-3: There is no indication in Fig. 5f which line is Pt/C and which line is sample.

Author Reply: We are very sorry to have caused your misunderstanding due to an improper figure caption. Actually, Fig. 5f presents the Chronopotentiometry curves of CoPGT at 10 mA cm^{-2} in 1.0 M KOH with 0.6 M NaCl , saturated NaCl , 0.6 M sea salt and saturated sea salt, respectively. There is no chronopotentiometry curve of $20\% \text{ Pt/C}$. According to your kind reminder, the figure caption has been corrected in the revised manuscript.

Comment 1-4: The XRD peaks in Fig. 3a are too weak, and some peaks, like the one around 40°, do not match to a standard card. The author should explain those peaks.

Author Reply: Thank you very much for your suggestion. The weak XRD peaks is mainly caused by the poor crystallinity of the CoP/rGO composite. The diffraction peak located at around 40° may can be identified as the (201) plane of Co₂P (JCPDS Card No. 89-3030) [Adv. Funct. Mater. **31**, 2107333 (2021)]. According to your kind suggestion, we have modified the description of XRD peaks and explained the peak around 40° in the revised manuscript as follows:

“A series of diffraction peaks located at 31.7°, 36.5°, 46.3°, 48.4° and 56.6° were observed, which are well indexed to the (011), (111), (112), (211) and (301) crystal planes of CoP phase (JCPDS Card No. 29-0497), further indicating that the β-Co(OH)₂ have been successfully converted to CoP. The diffraction peak marked with "*" near 40.6° is attributed to the (201) crystal face of Co₂P³², indicating the presence of a small amount of Co₂P.”

Fig. 3a XRD pattern of CoP/rGO.

Comment 1-5: In the stability test depicted in Figure 4d, the authors should explain why the potential of Pt/C decreased between 6 to 8 hours.

Author Reply: Thank you very much for your suggestion. In our opinion, the potential decrease of Pt/C between 6 to 8 hours is mainly caused by the accumulation and detachment of bubbles. In order to further verify this phenomenon, we repeated the relevant tests and prolonged the test time to 50 h (Fig. R2). It can be found that although the potential fluctuates during the test, the performance declined in the long-term run. Based on the valuable suggestions and comments from you and the Reviewer #4, Fig. 4d has been replaced by Fig. R2 in the revised manuscript.

Fig. R2 Chronopotentiometry curve of 20% Pt/C in 1.0 M KOH.

Response to Reviewer #3:

General Comment: The paper is an exciting topic, which has some interesting findings, and given the interest and importance of cobalt phosphides in water splitting, I am sympathetic with the view to publish. Focusing on the molecular dynamics (MD) component. The result that the surface of CoP excludes the halide is noteworthy and interesting. However, I consider that there are 2 major problems with the manuscript/SI that need addressing before the paper is accepted for publication with the MD included. Indeed, as a consequence of these, I do not find the MD results sufficiently convincing.

Author Reply: First of all, we sincerely appreciate your supporting and approval on this manuscript, which brings great encouragement and confidence to us. In particular, we are very grateful for your valuable comments and suggestions, which are definitely beneficial to improve the quality of our manuscript. The responses for your specific comments are listed below.

Comment 3-1: The first issue is that there is insufficient information to be able to reproduce the findings. The major omissions include a proper description of the force field and the surfaces. I appreciate the salt-water parameter set is referenced, but not the CoP and Co₃O₄ interaction parameters with themselves and critically, with water, chloride and alkali ions. What are the parameters used, what is the evidence that they are sensible? This is particularly important given that the metal phosphides are generally metallic. I presume, as "A few layers of CoP or Co₃O₄ were placed at the bottom of the box and fixed as the model of electrode surface", that the coordinates of the cobalt compounds were held fixed, if not then I would expect to see interaction parameters between all of the species.

Author Reply: Thank you very much for raising this issue as some details of the simulations had not been presented in the submitted manuscript and Supporting Information. With regard to the omission of proper descriptions of force fields and surfaces, we have added this point in the details of the theoretical simulations section and revised Supporting Information.

Interaction parameters for ions: The parameters of Lennard-Jones potential of Na^+ , K^+ and Cl^- are derived from reference [J. Phys. Chem. B **112**, 9020–9041 (2008)], as listed in Table R1, which well reproduce the solution properties and also the binding energies between ions and water, as well as the radii of the first hydration shells.

Interaction parameters for cobalt compounds: In this work, because there are not reasonable and direct potentials to describe the interaction of CoP or Co_3O_4 itself and also CoP and Co_3O_4 cannot deform or be destroyed in real catalysis process, the atomic coordinates of the cobalt compounds were held fixed in this work. Thus, the interactions of Co-P and Co-O in CoP and Co_3O_4 are not described. Nevertheless, the atomic interactions between CoP or Co_3O_4 and other compounds (ions, OH^- , H_2O) are described by Lennard-Jones potential and Coulomb interaction, and the corresponding parameters are listed in Table R1.

Interaction parameters for H_2O and OH^- : The H_2O molecule is described by SPC/E model [J. Phys. Chem. **91**, 6269–6271 (1987)]. The interaction within OH^- is derived from the CVFF force field [Proc. Natl. Acad. Sci. **85**, 5350–5354 (1988)]. The interactions between H_2O or OH^- and other compounds are described by Lennard-Jones potential and Coulomb interaction, and corresponding parameters are listed in Table R1. The geometric mixing rule is used to calculate the Lennard-Jones parameters for all species, and the cutoff

distance was 10 Å.

Table R1 LJ parameters for the saline ions, H₂O with substrates.

\		ϵ (kcal/mol)	σ (Å)	q (e)
Na ⁺		0.3526418 ¹	2.1595384928 ¹	+1
Cl ⁻		0.0127850 ¹	4.8304528498 ¹	-1
K ⁺		0.4297054 ¹	2.838403315 ¹	+1
OH ⁻	O	0.2280000124 ³	2.8597848722 ³	-1
	H	0 ³	0 ³	0
	H	0 ²	0 ²	0.4238
H ₂ O	H	0 ²	0 ²	0.4238
	O	0.1553 ²	3.166 ²	-0.8476
CoP	Co	0.0286 ⁴	1.2267 ⁴	2.43
	P	0.1999976833 ³	3.7417782334 ³	-2.43
Co ₃ O ₄	Co	0.0286 ⁴	1.2267 ⁴	2.43
	O	0.2280000124 ³	2.8597848722 ³	-1.8225

- [1] Nagashima, S. et al. Atomic-Level Observation of Electrochemical Platinum Dissolution and Redeposition. *Nano Lett.* **19**, 7000–7005 (2019).
- [2] Joung, I. S. & Cheatham, T. E. I. Determination of Alkali and Halide Monovalent Ion Parameters for Use in Explicitly Solvated Biomolecular Simulations. *J. Phys. Chem. B* **112**, 9020–9041 (2008).
- [3] Berendsen, H. J. C., Grigera, J. R. & Straatsma, T. P. The missing term in effective pair potentials. *J. Phys. Chem.* **91**, 6269–6271 (1987).
- [4] Maple, J. R., Dinur, U. & Hagler, A. T. Derivation of force fields for molecular mechanics and dynamics from ab initio energy surfaces. *Proc. Natl. Acad. Sci.* **85**, 5350–5354 (1988).

Fig. R3 Schematic diagram of a) CoP, b) Co₃O₄ substrate.

Based on the above description of interactions, Table R1 and Fig. R3 have been added as Supplementary Table 1 and Supplementary Fig. S3 in the revised Support Information, and the simulation details are further modified and improved in the revised manuscript as below:

“The interatomic interactions of catalyst with H₂O and various ions were described by Coulomb and Lennard-Jones potentials, and the corresponding parameters of Na⁺, Cl⁻, K⁺ were developed by In Suk Joung and Thomas E Cheatham⁵⁰, as listed in Supplementary Table 1. The interaction between H₂O molecules was described by SPC/E model⁵¹. The interaction within OH⁻ are derived from the CVFF force field and the interaction parameters of CoP and Co₃O₄ are referred from the previous report^{52,53}. All above details are shown in Supplementary Table 1. In addition, the detailed simulation models of both CoP and Co₃O₄ catalysts are shown in Supplementary Fig. 3.”

Comment 3-2: The other important piece of information missing is what surfaces were chosen? The ordering or otherwise of water on the surfaces (CoP and Co₃O₄) will depend on which surface structure, and arbitrarily selecting a disordered spinel surface for Co₃O₄ and a highly ordered CoP may increase the chance of getting the desired result. Also, why are those particular surfaces

chosen? Is there evidence that they are sensible surfaces? This becomes an even more critical assumption when the surface atoms held fixed.

In addition to answering the questions raised above, even if in the SI, please include input files for the 2 simulation cells, as this would also give the precise parameters chosen and help other researchers reproduce and develop the work.

Author Reply: According to the XRD results, lattice fringe calibration and electron diffraction analysis of the as-prepared catalyst in Fig. 2 and Fig. 3, we selected several very representative crystal faces (-102), (011) and (001) as the surfaces of the MD adsorption model, and then calculated the distribution of various ions on the CoP surface when the voltage was 0. As seen from Fig. R4 and Fig. 1b, when the surface of the MD adsorption model is taken as (001) crystal surface, there is not ion distribution near the catalyst surface, and the mass density of various ions near the surface is much lower than that of the other two surfaces. Therefore, we chose the (001) surface to continue the simulation at different voltages to explore whether the voltage can improve the ion adsorption or not. By comparison, the (001) surface of Co_3O_4 was also selected for further MD simulation to observe the ion distribution.

Fig. R4 Mass density of various ions versus the distance above the electrode surface with the presence of static external electric fields ($0 \text{ V } \text{\AA}^{-1}$) under different crystal surfaces (a) (011), (b) (-102) of CoP.

To make this manuscript readable and the results can be reproduced by others, the detailed simulation models of both CoP and Co₃O₄ have been added as Supplementary Fig. 3 (see the response of Comment 3-1) in the revised Supporting Information.

Comment 3-3: The second area of concern is the resulting water structure next to the CoP. However, I appreciate that I am inferring what I see from the projections. But it looks like the water is over bound, due to a poor representation of the electrostatics and short-range components leading to unphysical water ordering that then results in exclusion of the salt ions. Including rdfs for the Co and P with the oxygen and hydrogen of water would resolve this, so too the parameters for the model.

Author Reply: Thank you very much for this insightful comment. As explained above for atomic interactions with corresponding parameters in Table R1 (see the response of Comment 3-1), the short-range Lenard-Jones potential and electrostatic interaction between different species have been used to describe some similar solution systems or adsorption models. In this work, the simulated results are in agreement with experiment, indicating reasonable simulations. As you suggested, the radial distribution functions (rdfs, $g(r)$) for Co and P with oxygen and hydrogen of water are calculated. As is shown in Fig. R5, it can see that water molecules display different adsorption behaviours on CoP and Co₃O₄ surfaces. On CoP surface, the main peak of Co-O pair is much higher than that on Co₃O₄ surface, and its height increases as the voltage increases to 1 V/Å, similar to the P-O peak. This indicates that CoP surface has a stronger adsorption property to water molecules than Co₃O₄ surface, resulting in better H₂O catalysis action. In addition, the position of the main P-O peak (at about 2.9 Å) is further than the position of Co-O peak (at about 1.4

Å), which demonstrates that the electrostatic interaction plays an important role in water adsorption, rather than the over bound action.

Fig. R5 a-c) Radial distribution function of (001) surface of CoP substrate with H₂O under different voltages (0V/Å, -0.1V/Å, -1V/Å). d) Radial distribution function of (111) surface of CoP substrate with H₂O under 0V/Å.

According to the Fig.S2 in the supporting information of the literature searched [*Cell Rep. Phys. Sci.* **1**, 100136 (2020)], we can see that there are two absorption sites of hydrogen on the (111) surface of CoP and the distances between Co and H are 1.658 Å, 1.682 Å. Here, to further verify the accuracy of the electrostatics and short-range components, we also simulated the adsorption of water on the (111) surface of CoP and obtained the radial distribution functions (rdfs), shown in Fig. R5. From rdf of Co and P with oxygen and hydrogen of water in Fig. R5(d), we can find that the peak position between Co and H are at about 1.65833 Å, which is completely same to the reference value of 1.658 Å and also has a very small difference with 1.682 Å.

We believe that the error range is acceptable, meaning that the parameters are reasonable. Therefore, we hold the opinion that the water next to the surface is not over bound.

Comment 3-4: Finally, I was confused about the periodicity used, was it 2D or 3D? Particularly, if 3D, rather than just stating that the field is $-1 \text{ V}/\text{\AA}$, on your structural figure, identify which is the negative and positive sides.

Author Reply: Thank you for your kind suggestion. In the case presented herein, the periodicity of the whole model is 3D. In order to avoid causing the readers misunderstanding, we have made corresponding modifications to the caption of Fig. 1a as follows:

Fig. 1 MD simulations. a Equilibrium configuration of electrolyte system (1.0 M KOH + 0.6 M NaCl) above the electrode surface of CoP with the presence of static external electric fields ($-1.0 \text{ V } \text{\AA}^{-1}$), viewed from XZ cross-section.

In the part of simulations, the electric field was exerted by the command in Lammmps (that is the simulation software), namely “fix ID group-ID style ex ey ez keyword value”. The constant electric field was performed only in Z direction, so both ex and ey (electric field intensity in X and Y directions) were set as 0. Here, the external electric field was exerted by adding a constant force to each charged atom in solution, according to the equation $F = qE$ where q is the charge and E is the electric field. A negative electric field intensity indicates that the direction of the electric field is along -Z direction (from top to bottom of model), and vice versa. This method of applying an electric field does not strictly distinguish positive and negative poles, but represents the gradient voltage with 0.1 or 1.0 V per unit separation (in angstrom, \AA) from 0 to another value, meaning that the lower the position is, the lower the electric potential. This method has been widely used to exert electric field in many studies [Chem.

Eng. J. **363**, 278–284 (2019). *Chem. Eng. J.* **418**, 129391 (2021). *Angew. Chem. Int. Ed.* **60**, 22740–22744 (2021)].

Response to Reviewer#4:

General Comment: In this study, the authors provide a new CoP/rGO@Ti electrode for the hydrogen evolution reaction in seawater electrolysis. The HER activity of CoP/rGO@Ti electrode in alkaline seawater electrolyte is excellent due to the highly effective repulsive Cl^- intrinsic characteristic of CoP. The results are interesting and may contribute well to the electrocatalytic community in producing hydrogen using the seawater-based electrolyte.

Author Reply: First of all, thank you very much for your support and affirmation of this manuscript, especially for your comprehensive interpretation of this work. Definitely, we also sincerely appreciate all your valuable comments and suggestions, which greatly help us to improve the quality of our current manuscript. According to your helpful suggestions, necessary revisions have been made in the revised manuscript. The responses to your specific comments are listed below.

Comment 4-1: Please add more detailed information about the materials used in the experiment. In addition, what is the purity of KOH?

Author Reply: We are grateful for this suggestion. In this work, the purity of the KOH used in the experiment was 99.99%. More detailed information about the materials used in the experiment has also been added in the revised manuscript as follows:

“**Chemicals:** Ammonia water (25%), graphene oxide (10 mg mL^{-1}), cobaltous nitrate hexahydrate ($\text{Co}(\text{NO}_3)_2 \cdot 6\text{H}_2\text{O}$) and hydrazine monohydrate ($\text{N}_2\text{H}_4 \cdot \text{H}_2\text{O}$, 85%) were supplied by Sinopharm Chemical Reagent Co., Ltd. (China). Potassium hydroxide (KOH, Aladdin, 99.99%), Pt/C (20 wt% Pt), sodium hypophosphite (NaH_2PO_2), Nafion (5 wt%), Sodium chloride (NaCl) and ethanol ($\text{C}_2\text{H}_5\text{OH}$) were purchased from Shanghai Aladdin Bio-Chem

Technology Co., Ltd. (China). Titanium foil obtained from CeTech Co., Ltd. (China). All reagents were used as received without further purification. The water was purified through a Millipore system throughout all experiments.”

Comment 4-2: For comparison of the HER activity of the prepared catalyst with the 20wt% Pt/C, the 20wt% Pt/C should be deposited on TF (1.5 mg/cm²) to record its HER activity. I believe that the low loading of Pt/C tested in the RDE system leads to low HER performance at a high current density.

Author Reply: Thank you very much for raising this issue. According to your suggestion, 20wt% Pt/C with a mass density of 1.5 mg/cm² was dripped onto the surface of the TF to investigate its HER activity. Unfortunately, we found that Pt/C powder is prone to shedding during electrolysis of water, and it is due to this that the Pt/C powder does not bind well with TF, leading to a higher impedance of charge transfer at the interface, which results in poor HER performance (Fig. R6).

Fig. R6 LSV curve of 20% Pt/C@TF.

Comment 4-3: The morphology of Co₃O₄GT and CoPT catalysts should be proved by SEM images to compare with that of CoGPT.

Author Reply: Thank you very much for your valuable suggestion. The SEM images of $\text{Co}_3\text{O}_4\text{GT}$ and CoPT have been added as the Supplementary Fig. 8 and Fig. 9 in the revised Supporting Information. The corresponding description of the two figures have also been added in the revised manuscript as follows:

“The $\text{Co}_3\text{O}_4\text{GT}$ and CoPT catalysts were prepared for comparison (Supplementary Fig.8 and 9). Both of them present the similar frame structure with CoPGT . The corresponding elemental mapping images show the $\text{Co}_3\text{O}_4\text{GT}$ with uniform distribution of Co, P and C elements and the CoPT with uniform distribution of Co and P, which are beneficial to exert their catalytic performance.”

Supplementary Fig. 8 a) SEM image and the corresponding elemental mapping images b) Co, c) O, d) C elements of $\text{Co}_3\text{O}_4\text{GT}$.

Supplementary Fig. 9 a-b) SEM images and the corresponding elemental mapping images c) Co, d) P elements of CoPT.

Comment 4-4: Please insert the enlarged Figure 4a to see the values of potential at a current density of -10 mA cm^{-2} , which values were mentioned in the manuscript.

Author Reply: We appreciate your kind suggestion. The enlarged Figure 4a with the marked potential values at a current density of -10 mA cm^{-2} has been inserted as follows in the revised manuscript.

Fig. 4a LSV curves of CoPGT, CoPT, CoP, $\text{Co}_3\text{O}_4\text{GT}$, rGO, Ti and 20% Pt/C for the HER in 1.0 M KOH, inset: the detailed at -10 mA cm^{-2} .

Comment 4-5: The time for the stability test is 10 h to be short and not easy to say the stability of catalysts.

Author Reply: Thank you very much for this comment. In order to better evaluate the stability of the catalysts, the test time of the CoPGT and 20% Pt/C catalysts was prolonged to 50 h (Fig. R7) and the CoPGT at 10 mA cm⁻² in 1.0 M KOH with different salt concentrations was prolonged to 60 h (Fig. R8). It can be seen that the catalyst perform keeps very stable without notable degradation during 50 h electrolysis at 10 mA cm⁻² (Fig. R7), indicating the excellent stability of CoPGT electrocatalyst. Especially, even the current density increased to 200 mA cm⁻², the performance of CoPGT is still far more stable than that of 20% Pt/C. For the CoPGT catalyst in 1.0 M KOH with different salt concentrations (Fig. R8), it still performed well without significant loss in simulated and real seawater during the 60 h CP test, further demonstrating its good structure stability. Fig. 4d and Fig. 5f have been respectively replaced by Fig. R7 and Fig. R8 in the revised manuscript, and the related description has been revised.

Fig. R7 Chronopotentiometry curves of CoPGT and 20% Pt/C in 1.0 M KOH.

Fig. R8 Chronopotentiometry curves of CoPGT at 10 mA cm^{-2} in 1.0 M KOH with 0.6 M NaCl , saturated NaCl , 0.6 M sea salt and saturated sea salt, respectively.

Comment 4-6: What is the potential range for cyclic stability? It is an important parameter for readers.

Author Reply: Thank you very much for pointing out this negligence. The cyclic stability of CoPGT was tested by repeating LSV running for 1000 cycles at a scan rate of 100 mV s^{-1} with the potential range between -0.80 and -1.20 V (vs RHE), which has been added in the Electrochemical measurements section of the revised manuscript.

Comment 4-7: SEM image of CoPGT was shown in Supplementary Fig. 11 to see the unchanged morphology of CoPGT after the stability test. It would be better to show the TEM image of CoPGT for clarifying CoPGT morphology.

Author Reply: We really appreciate your valuable advice. The TEM image of CoPGT after the stability test has been provided as Supplementary Fig. 16 in the revised Supporting Information. It can be seen that the CoPGT after the

stability test still keep a sheet-like morphology with uniform distribution of Co, P and C elements. The corresponding description has also been added in the revised manuscript.

Supplementary Fig. 16 Morphology characterization. a) TEM image and the corresponding elemental mapping images b) Co, c) P, d) C elements of CoP/rGO after 60 h CP tests in 1.0 M KOH.

Comment 4-8: Why the values of overpotential difference against salt concentration gradient for CoPGT and CoPT are different as seen in Fig. 5e, even though they have the same mechanism with the highly effective repulsive Cl⁻ intrinsic characteristic of CoP? Please the authors give comments.

Author Reply: As is well-known to all, the overpotential of the electrocatalyst is influenced by many factors, including the catalyst microstructure and charge transfer resistance (R_{ct}) [ACS Nano 12 (10), 9635-9638 (2018); Joule 2 (6), 1024-1027 (2018); J. Electroanal. Chem. 666, 89 (2012); Nat. Mater. 15 (9) 1003-1009 (2018)]. In this work, although both CoPGT and CoPT can effectively repulse Cl⁻ via CoP, their microstructure and R_{ct} are different due to the addition of rGO. The nitrogen adsorption-desorption test results show that the surface area

of CoP/rGO is higher than double that of pure CoP powder (Supplementary Fig. 10), which is attributed to the avoiding agglomeration by the addition of rGO. The Barrett-Joyner-Halenda (BJH) pore-size distribution curve of CoP/rGO shows a higher pore volume, indicating a higher electrochemical surface area, which is beneficial to enhance the active surface area of catalyst. In addition, the CoP/rGO catalyst presented a smaller R_{ct} value than CoP (Supplementary Fig. 20), implying a faster charge transfer, which is attributed to the excellent electrical conductivity of rGO. Therefore, CoPGT possesses better characteristics to resist salinity change, resulting the overpotential difference of CoPGT is smaller than that of CoPT.

Comment 4-9: Supplementary Fig. 9 and 15 are not mentioned in the manuscript.

Author Reply: Sorry for our careless and thank you very much for your kind reminder. Supplementary Fig. 9 and 15 and the corresponding description have been added in the revised manuscript. It should be noted that the serial number of Supplementary Fig. 9 and 15 has been adjusted to Supplementary Fig. 12 and 18 in the revised manuscript due to the addition of new figures.

The description of Supplementary Fig. 12 is as follow:

“The XRD pattern and XPS spectra of $\text{Co}_3\text{O}_4\text{GT}$ in Supplementary Fig. 12 confirm that the oxidized Co species is attributed to Co_3O_4 .”

The description of Supplementary Fig. 18 is as follow:

“In order to further demonstrate the salinity tolerance of CoP, the electrocatalytic properties of CoP catalyst ink was also studied in the electrolyte with chloride sodium (NaCl) concentration from 0 M to saturation. As shown

in Supplementary Fig. 18, the catalytic performance of CoP catalyst ink has no obvious attenuation with increasing concentration of NaCl as expected.”

Comment 4-10: Since values of electrochemically active surface area (ECSA), charge transfer resistance (R_{ct}), and turnover frequency (TOF) are very important parameters to evaluate the catalytic activity of HER catalysts. Please provide them.

Author Reply: We sincerely appreciate for this valuable suggestion. Electrochemically active surface area (ECSA), charge transfer resistance (R_{ct}), and turnover frequency (TOF) have been added in the revised Supporting Information as Supplementary Fig. 19 and Fig. 20. Meanwhile, the related description has also been supplemented and expanded in the revised manuscript.

The following text has been added in Electrochemical measurements section of the revised manuscript:

“The electrochemical double-layer capacitance (C_{dl}) of catalysts were evaluated by using cyclic voltammetry in a non-Faradaic region from -0.40 to -0.50 V vs. Hg/HgO at different scan rates ranging from 10 to 60 mV s⁻¹. The C_{dl} equals to the resulting linear slope of the difference between the half of the current density of anodic and cathodic at -0.45 V vs. Hg/HgO versus scan rate. The turnover frequency (TOF) value is estimated by the following equation: $TOF = \frac{j \times S}{Z \times N \times F}$. Here, j is the current density, S is the area of the electrode, Z represents the electrons transfer number in HER which is 2, F is the Faraday constant (96485.3 C mol⁻¹), and N is the number of sites of the active materials. The content of Co element was 21.82 wt%, as measured by inductively coupled plasma mass spectrometry (ICP-MS).”

Supplementary Fig. 19 Determination of Cdl. CV curves of a) CoP, b) CoP/rGO with various scan rates of 10, 20, 30, 40, 50 and 60 mV s^{-1} in the non-faradaic region. b) The capacitive current densities as a function of scan rate for the catalysts.

Supplementary Fig. 20 EIS Nyquist plots of CoP/rGO and CoP.

The related discussion has been added in Results Section of the revised manuscript as follows:

“The TOF of CoP/rGO is 0.0239 s^{-1} , indicating a high intrinsic activity. To figure out the origin of the enhanced activity of CoP/rGO electrocatalyst, the double-layer capacitances (C_{dl}) measurements⁴⁷ were carried out to evaluate the electrochemical active surface areas (Supplementary Fig. 19). The CoP/rGO showed a higher C_{dl} of 2.33 mF cm^{-2} than CoP, indicating more electroactive surface exposed in CoP/rGO compare with pure CoP. The electrochemical

impedance spectroscopy (EIS) was performed to deeply study the charge-transfer mechanism and the resulted Nyquist plots were shown in Supplementary Fig. 20. Apparently, the CoP/rGO catalyst presented a smaller R_{ct} than CoP, implying a faster charge transfer."

Comment 4-11: To exclude the possibility of side reactions during the HER under half-cell measurement, the full cell test should be investigated for CoPGT. H_2 product should be collected and calculated the Faradaic efficiency during the full-cell operation of water splitting.

Author Reply: Thank you very much for this worthy suggestion. The Faradaic efficiency during the full-cell operation of water splitting was calculated and the result has been added in the revised Supporting Information as Supplementary Fig. 14. The related description also has been added in the revised manuscript.

Supplementary Fig. 14. Faradic efficiency measurements. The amount of H_2 experimentally measured and theoretically calculated versus time for CoP/rGO in 1.0 M KOH.

The following text has been added in Electrochemical measurements section:

“The Faraday efficiency (ϵ) of HER was determined by measuring the amount of H₂ produced. The Faraday efficiency (FE) can be calculated as follows: $FE_{H_2} = \frac{n_{H_2} \times Z \times F}{Q_s}$. Where n_{H_2} is the amount of the hydrogen, which was measured using gas chromatography (8860 GC). The catalytic gas is collected for 90 min and measuring the amount of H₂ produced every 10 min. Z is the number of transferred electrons (Z=2 in this work); F is the Faraday constant, Q_s is the total charge in the reaction for 60 min, which was calculated by the integration of Chronoamperometry curve. The Faradaic efficiency tests were performed in a two-electrode system using an electrochemical workstation (CHI 660E). The CoPGT catalyst and Pt foil were used as the counter electrode (CE) and reference electrode (RE), respectively.”

The related discussion is added as follow in Results section:

“In addition, as shown in Supplementary Fig. 14, the faradic efficiency of CoPGT catalyst for HER is estimated to be close to 100%, indicating that almost all electrons are utilized for producing hydrogen.”

List of Revisions

In the Manuscript

1. Page 7: the text “The $\text{Co}_3\text{O}_4\text{GT}$ and CoPT catalysts were prepared.....which are beneficial to exert their catalytic performance” is added.
2. Page 8: the text “A series of diffraction peaks located at 31.7° , 36.5° , 46.3° , 48.4° , 52.3° and 56.6° were observed.....further indicating that the $\beta\text{-Co(OH)}_2$ have been successfully converted to CoP ” is replaced by the text “A series of diffraction peaks located at 31.7° , 36.5° , 46.3° , 48.4° and 56.6° were observed.....indicating the presence of a small amount of Co_2P ”.
3. Page 8-9: the text “The peaks at binding energies of 779.6 eV and 795.1 eV are assigned to Co with a partial positive charge for CoP^{39}while the peak located at 133.5 eV correspond to the phosphate due to surface oxidation^{42,43}.” is added.
4. Page 10: the text “In addition, as shown in Supplementary Fig. 14, the faradic efficiency of CoPGT catalyst for HER is estimated to be close to 100%, indicating that almost all electrons are utilized for producing hydrogen.” is added.
5. Page 11: the text “the TEM image (Supplementary Fig. 16)” is added.
6. Page 11-12: the text “In order to further demonstrate the salinity tolerance of CoP the CoP/rGO catalyst presented a smaller R_{ct} value than CoP , implying faster charge transfer.” is added.
7. Page 12: the word “10 h” is replaced by “60 h”.
8. Page 14: the word “Chemicals” and the text “Ammonia water (25%), graphene oxide (10 mg mL^{-1}).....The water was purified through a Millipore system throughout all experiments.” is added.

9. Page 18: the text “The stability of CoPGT was tested by repeating LSV running for 1000 cycles.....as measured by inductively coupled plasma mass spectrometry (ICP-MS).” is added.
10. Page 19: the text “The Faraday efficiency (ϵ) of HER was determined by measuring the amount of H₂ produced.....The CoPGT catalyst and Pt foil were used as the counter electrode (CE) and reference electrode (RE), respectively.” is added.
11. Page 19: the word “10 hours” is replaced by “60 hours”.
12. Page 20: the text “as listed in Supplementary Table 1.....the detailed simulation models of both CoP and Co₃O₄ catalysts are shown in Supplementary Fig. 3.” is added.
13. Page 25: references [47], [50, [52-53] are added, and the other reference number is adjusted accordingly.
14. Page 28: the caption “Snapshots of molecular dynamics simulations of electrolyte systems (1.0 M KOH + 0.6 M NaCl) above the electrode surface of CoP with the presence of static external electric fields (-1.0 V \AA^{-1})” is replaced by “Equilibrium configuration of electrolyte system (1.0 M KOH + 0.6 M NaCl) above the electrode surface of CoP with the presence of static external electric fields (-1.0 V \AA^{-1}), viewed from XZ cross-section”.
15. Page 30: Fig. 3a-3c are redrawn.
16. Page 31: Fig. 4a is redrawn and Fig. 4d is revised. In addition, the caption “inset: the detailed at -10 mA cm^{-2} ” is added.
17. Page 32: the caption of Fig. 5f was replaced by “Chronopotentiometry curves of CoPGT at 10 mA cm^{-2} in 1.0 M KOH with 0.6 M NaCl, saturated NaCl, 0.6 M sea salt and saturated sea salt, respectively.”.

In the Support Information

1. Page 2: the caption of **Supplementary Fig. 2a** is replaced by “**Equilibrium configuration of electrolyte system (1.0 M KOH + 0.6 M NaCl) above the electrode surface of Co₃O₄ with the presence of static external electric fields (-1.0 V \AA^{-1}), viewed from XZ cross-section**”.
2. Page 3: the **Supplementary Fig. 3** is added.
3. Page 8: the **Supplementary Fig. 8** is added.
4. Page 9: the **Supplementary Fig. 9** is added.
5. Page 14: the **Supplementary Fig. 14** is added.
6. Page 15: the **Supplementary Fig. 15** is added.
7. Page 16: the **Supplementary Fig. 16** is added.
8. Page 19: the **Supplementary Fig. 19** is added.
9. Page 20: the **Supplementary Fig. 20** is added.
10. Page 23: the **Supplementary Table 1** is added.
11. Page 25: references [1-4] are added, and the other reference number is adjusted accordingly.

REVIEWER COMMENTS

Reviewer #1 (Remarks to the Author):

Although the authors have answered most of the questions. The following issues still needed to be resolved.

1. Regarding the ADT in Figure 4c, I believe that for the alkaline hydrogen evolution reaction, the stability test with 1000 cycles lacks strong persuasiveness. Based on recent literature about HER, accelerated stability tests typically involve more than 10,000 cycles. Therefore, I suggest that the authors increase the number of stability test cycles (preferably above 10,000 cycles) to enhance the persuasiveness of the data.
2. According to the stability test Figure 5f, a CA test with 0.6 M sea salt and saturated sea salt was added to assess stability in a seawater environment. However, there was no corresponding LSV test conducted in a sea salt solution. The author should explain the reason for this omission or consider conducting LSV tests in a sea salt solution.
3. To make the experimental system more comprehensive, standard reference samples like Pt/C should also undergo performance testing in solutions containing NaCl and sea salt. This will better highlight the superiority of CoP in harsh seawater environments and enhance the rigor and persuasiveness of the experiments.

Reviewer #3 (Remarks to the Author):

Manuscript number: NCOMMS-23-19305A

Title: "Corrosion-resistant Cobalt Phosphide Electrocatalysts for Salinity Tolerance Hydrogen Evolution"

I have reviewed their rebuttal and looked again at the manuscript. I consider that in terms of the molecular simulation component of the manuscript, that they have addressed all the points raised. And while I might have been tempted to use slightly different potential model parameters (for the CoP), and other surface structures, I don't see anything wrong with the assumptions made, especially as there is sufficient information to repeat their work.

Therefore, I would recommend accepting.

Reviewer #4 (Remarks to the Author):

The authors have revised the manuscript very well. The manuscript is greatly improved, especially in the aspect of material characterizations and electrochemical performance. Specifically, the ECSA, TOF, Rct values, and the stability of the catalyst were investigated and provided in detail. The HER catalyst was applied to the full cell and showed good results consistent with values obtained from the half-cell, suggesting this CoP/rGO@Ti electrode is promising for alkaline seawater electrolysis in hydrogen production. This research is interesting and could attract wide attention from scientists in the field of materials and catalysis. Thus, I suggest the publication of it in Nature Communication in this present form.

We sincerely appreciate all of these valuable suggestions from the Reviewer #1, which would undoubtedly help us to further improve the quality of our current manuscript. Following are our responses to these suggestions.

Content:

To Reviewer #1:.....	2
List of Revisions	6

To Reviewer #1:

General Comment: Although the authors have answered most of the questions. The following issues still needed to be resolved.

Author Reply: Thank you very much for your professional and valuable suggestions, which are definitely beneficial to further improve the quality of our manuscript. According to these precious advices, corresponding data have been added in the revised manuscript. The responses for your specific comments are listed below.

Comment 1-1: Regarding the ADT in Figure 4c, I believe that for the alkaline hydrogen evolution reaction, the stability test with 1000 cycles lacks strong persuasiveness. Based on recent literature about HER, accelerated stability tests typically involve more than 10,000 cycles. Therefore, I suggest that the authors increase the number of stability test cycles (preferably above 10,000 cycles) to enhance the persuasiveness of the data.

Author Reply: We sincerely appreciate this valuable suggestion. The stability test of the CoPGT catalyst was prolonged to 10,000 cycles, and the test result is presented as Fig. R1. It can be seen that the polarization curve of the CoPGT catalyst shows no significant change even after 10,000 cycles and the overpotential is only increased by 10 mV at the current density of 10 mA cm⁻², indicating its stable catalytic performance. Fig. 4c has been replaced by Fig. R1 in the revised manuscript.

Fig. R1 Polarization curves of CoPGT initially and after 10000 cycles at a scan rate of 1 mV s⁻¹ for HER in 1.0 M KOH, inset: overall view of LSV curves.

Comment 1-2: According to the stability test Figure 5f, a CA test with 0.6 M sea salt and saturated sea salt was added to assess stability in a seawater environment. However, there was no corresponding LSV test conducted in a sea salt solution. The author should explain the reason for this omission or consider conducting LSV tests in a sea salt solution.

Author Reply: Thank you very much for this professional suggestion. The LSV tests of CoPGT at 10 mA cm^{-2} in 1.0 M KOH containing different concentrations of sea salt and NaCl have been conducted, and the corresponding test results are shown as Fig. R2. It can be seen that no significant degradation can be found in the polarization curves after 10000 LSV cycles no matter in sea salt solutions or in NaCl solutions, demonstrating the excellent salinity resistance of the CoPGT catalyst. Fig. R2 has been added in the revised Supporting Information as Supplementary Fig. 24. The related description has also been added in the revised manuscript.

Fig. R2 Salinity resistance performance. Polarization curves of CoPGT initially and after 10000 cycles at a scan rate of 1 mV s^{-1} for HER in 1.0 M KOH with a) 0.6 M sea salt and b) saturated sea salt, c) 0.6 M NaCl, d) saturated NaCl, respectively.

Comment 1-3: To make the experimental system more comprehensive, standard reference samples like Pt/C should also undergo performance testing in solutions containing NaCl and sea salt. This will better highlight the superiority of CoP in harsh seawater environments and enhance the rigor and persuasiveness of the experiments.

Author Reply: Thank you very much for this precious suggestion. The CA and LSV tests of 20% Pt/C have been supplemented to make the experimental system more comprehensive. Fig. R3 is the chronopotentiometry curves of 20% Pt/C at 10 mA cm^{-2} in 1.0 M KOH containing different concentrations of sea salt and NaCl. It can be seen that the overpotentials of 20% Pt/C catalyst gradually increases with the extension of test time, while the overpotentials of CoPGT remain stable, indicating its superior stability. Meanwhile, the CoPGT electrocatalyst exhibits a negligible overpotential variation after 10000 times continuous LSV scanning at 10 mA cm^{-2} no matter in NaCl solutions or sea salt solutions (Fig. R2), further demonstrating its excellent salinity resistance. In contrast, the overpotentials of the 20% Pt/C catalyst increased significantly after 10000 LSV cycles in 1.0 M KOH containing different concentrations of sea salt and NaCl (Fig. R4), suggesting its poor stability in harsh seawater environments. Fig. R3 and R4 have been added as Supplementary Fig. 22 and Fig. 25 in the revised Supporting Information. Meanwhile, the related description has also been supplemented and expanded in the revised manuscript.

Fig. R3 Electrochemical HER performance measurements. Chronopotentiometry curves of a) 20% Pt/C, b) CoPGT at 10 mA cm⁻² in 1.0 M KOH with 0.6 M NaCl, saturated NaCl, 0.6 M sea salt and saturated sea salt, respectively.

Fig. R4 Salinity resistance performance. Polarization curves of 20% Pt/C initially and after 10000 cycles at a scan rate of 1 mV s⁻¹ for HER in 1.0 M KOH with a) 0.6 M sea salt and b) saturated sea salt, c) 0.6 M NaCl, d) saturated NaCl, respectively.

List of Revisions

In the Manuscript

1. Page 10: the word “1000 cycles” is replaced by “10000 cycles”; the text “no significant change can be found in the polarization curves after 1000 LSV cycles, indicating its stable catalytic performance.” is replaced by the text “the polarization curve of the CoPGT catalyst.....indicating its stable catalytic performance.”
2. Page 12: the text “which is much better than 20% Pt/C.....suggesting its poor stability in harsh seawater environments.” is added.
3. Page 13: the text “Meanwhile, after continuous LSV scanning for 1000 cycle..... in electrolyte (Supplementary Fig. 22).” is deleted.
4. Page 18: the word “20% Pt/C” is added.
5. Page 32: Fig. 4c is redrawn. In addition, the word “1000 cycles” is replaced by “10000 cycles”, and the text “inset: overall view of LSV curves” is added.

In the Support Information

1. Page 22: the Supplementary Fig. 22 is added.
2. Page 24: the Supplementary Fig. 24 is added.
3. Page 25: the Supplementary Fig. 25 is added.